# MXSens: Sensitivity-Aware Mixed-Precision Quantization for Efficient LLM Inference

## Abstract

4-bit quantization enables efficient LLM inference, but suffers from significant accuracy degradation due to outliers. Prior work addresses this problem via data rotation or mixed-precision integer quantization, but often relies on software-managed scaling and frequent dequantization, incurring substantial overhead. Microscaling formats, such as MXINT, eliminate these inefficiencies by encoding scales in hardware, yet remain incompatible with rotation-based methods. Our analysis reveals that outliers vary in severity, from rare extremes to frequent mild deviations, and that quantization sensitivity is unevenly distributed across layers and columns. These insights motivate a fine-grained, sensitivity-guided approach. We introduce MXSens, a training-free method that assigns mixed mantissa bitwidths (4/6/8) based on column- and layer-wise sensitivity, naturally leveraging the block-wise structure of MXINT. MXSens outperforms state-of-the-art quantization methods across a range of models and tasks. Under the W4A4KV4 setting, MXSens achieves perplexities of 3.77 and 7.63 on LLaMA-2-70B and LLaMA-3-8B, respectively, substantially improving over existing baselines on WikiText-2. Our work establishes a new balance between accuracy and resource efficiency for LLM quantization.

## 1 Introduction

Quantization has emerged as a critical technique for enabling time- and energy-efficient inference in large language models (LLMs), significantly reducing memory footprint and accelerating execution by using low-precision numerical formats. Despite its effectiveness, LLM quantization remains challenging due to the presence of systematic outliers in activations (Dettmers et al., 2022). These high-magnitude values distort the distribution within quantization blocks and complicate the selection of appropriate scaling factors–particularly in low-bitwidth regimes such as 4-bit quantization. Effectively addressing these outliers is therefore essential for maintaining model accuracy under aggressive quantization.

Recent methods tackle this problem in two main ways: (1) reshaping the tensor distributions using rotation matrices, as in QuaRot (Ashkboos et al., 2024b), RRS (Yi et al., 2025), and SpinQuant (Liu et al., 2024), or (2) applying mixed-precision quantization, as done in Atom (Zhao et al., 2024), QoQ (Lin et al., 2024b), and QUIK (Ashkboos et al., 2024a). All of these approaches use INT4 as the primary number format, but to maintain accuracy, they rely on sub-channel or per-group quantization with software-handled FP32 scaling. This approach requires frequent dequantization operations to and from FP32 during inference, which can incur up to 90% dequantization overhead relative to pure INT4 baselines (Lin et al., 2024b), significantly undermining the benefits of low-bit quantization.

Microscaling formats eliminate this overhead by embedding power-of-two scale factors directly into the numeric representation (Rouhani et al., 2023a;b). Instead of relying on software-handled FP32 scales, microscaling formats embed a shared power-of-two scale per small block (e.g., 32 values), enabling low-bit inference using fixed-point MACs in the case of MXINT, or low-bit floating-point in the case of MXFP. This

hardware-friendly design maps naturally to modern AI accelerators and is already supported in practice, including on NVIDIA Blackwell GPUs (Nvidia, 2024) and Qualcomm Cloud AI 100 (Qualcomm, 2025). However, microscaling formats are inherently incompatible with rotation-based methods for outlier handling. As shown in AMXFP (Lee et al., 2025), applying rotations on MXFP/MXINT tensors introduces group-wise asymmetry, which degrades quantization quality and cancels out the accuracy benefits of fine-grained scaling. This incompatibility motivates a new direction: instead of modifying the tensor distributions via rotation, we focus on modifying precision, guided by the native structure of microscaling formats.

To enable precision-based outlier mitigation, we analyze the quantization sensitivity of LLMs and find that it varies widely across the model. Consistent with recent studies that categorize outliers into different types (Lin et al., 2024a; Yi et al., 2025), our analysis reveals a broader spectrum of activation sensitivity. Outliers vary in magnitude, ranging from rare, extremely high values to more moderate deviations, which we refer to as *mild outliers* and still cause significant quantization error under 4-bit formats. These sensitivities are not uniformly distributed: some layers and columns are more affected than others. This variation in sensitivity motivates fine-grained precision allocation across weights, activations, and KV-cache. However, existing mixed-precision methods typically rely on static or heuristic bitwidth assignment, ignoring these differences and leading to suboptimal use of precision. Based on this analysis, we conclude that more flexible bitwidth assignment is needed. Microscaling formats, especially MXINT, provide a natural fit for such adaptation due to their block-wise layout and hardware compatibility (Zhang et al., 2022; Noh et al., 2023).

We propose a novel outlier-aware quantization method MXSens, leveraging column- and layer-wise sensitivity to minimize accuracy loss while maintaining memory and compute efficiency without additional training. Building on recent advances in microscaling formats, MXSens is the next step toward practical, high-accuracy MXINT4 deployment: our sensitivity analysis shows that selective precision allocation is both necessary for accuracy and naturally supported by MXINT's block-wise structure. Our contributions are as follows:

- We introduce a sensitivity-guided *triplet quantization* method that uses three distinct bitwidths (e.g., 4, 6, and 8 bits) for weights and activations, achieving state-of-the-art accuracy at comparable compression rates without modifying the model architecture or requiring retraining or fine-tuning.

- We adopt the MXINT numeric format for our quantization method, leveraging its block-wise structure and shared exponent to enable efficient mixed-precision implementation.

- MXSens outperforms SOTA quantization methods across a range of models and tasks. In particular, under the W4A4KV4 setting, MXSens achieves perplexities of 3.77 and 7.63 on LLaMA-2-70B and LLaMA-3-8B, respectively, on WikiText-2, substantially improving over existing baselines.

## 2 BACKGROUND

Microscaling formats (Rouhani et al., 2023c), such as MXFP and MXINT, have recently emerged as promising numeric representations for low-bit inference. They strike a balance between dynamic range and hardware efficiency by applying a shared power-of-two scale to fixed-size blocks of values (typically 32), encoded as a floating-point-style exponent. In these formats, MXFP stores each value as a low-bit mantissa and exponent, while MXINT simplifies further by using fixed-point integers within each block. The latter enables efficient processing using standard integer multiply–accumulate units. Microscaling formats are already seeing adoption in hardware, including NVIDIA's Blackwell GPUs (Nvidia, 2024) and Qualcomm's Cloud AI 100 (Qualcomm, 2025), both of which support block-level scaling and multiple precision modes.

Despite this progress, quantization at 4-bit precision still suffers from substantial accuracy loss, primarily due to outliers, which are defined as a small number of disproportionately large values in the tensor (Timkey & Van Schijndel, 2021; Bondarenko et al., 2021; Dettmers et al., 2022; Wei et al., 2022; Puccetti et al., 2022). These values dominate the shared scale of their block, compressing the representational range available for the

remaining elements. Two main approaches have emerged as a solution for outliers: rotation-based and mixed-precision quantization. Rotation methods aim to reshape the input distribution before quantization (Ashkboos et al., 2024b; Yi et al., 2025; Lin et al., 2024a; Liu et al., 2024). However, such transformations often introduce asymmetry that disrupts the block structure required by microscaling formats, as shown in AMXFP (Lee et al., 2025). Rotation methods also face deployment challenges. For example, SpinQuant's rotations are particularly limited on models such as Gemma-2 (Mesnard et al., 2024), which feature both pre-norm and post-norm structures (Liu et al., 2024). In addition, these methods typically leave certain components—most notably the Softmax output vectors—unquantized.

In contrast, mixed-precision quantization is naturally aligned with the structure of microscaling formats. Instead of reshaping distributions, it assigns higher precision to blocks or channels with high quantization sensitivity (Zhao et al., 2024; Ashkboos et al., 2024a; Lin et al., 2024b; Liu et al., 2025; Ramachandran et al., 2025). This approach is especially effective in MXINT, which uses fixed-point arithmetic and can vary mantissa bitwidths across blocks with minimal hardware changes. Our sensitivity analysis shows that quantization error tends to concentrate in a small subset of blocks or columns, meaning that selectively increasing precision in just those areas can significantly boost accuracy. This makes mixed-precision MXINT a practical and scalable approach to high-accuracy low-bit inference, preserving the compactness and compatibility of MX formats while addressing the limitations of uniform 4-bit quantization.

## 3 SENSITIVITY METRICS FOR MIXED-PRECISION

Quantizing specific components of a model, such as certain columns and layers, can have a disproportionate impact on overall model accuracy. While assigning uniform bitwidth across all layers and columns offers simplicity, it overlooks the substantial variation in quantization sensitivity observed across different parts of LLMs. As models scale and vary in architecture, the magnitude and distribution of activation outliers differ significantly, resulting in substantial variability in quantization sensitivity across columns and layers. Identifying and understanding these sensitive components is crucial as it enables targeted allocation of higher precision, significantly improving the trade-off between accuracy and hardware overhead. In this section, we present a systematic method for evaluating column- and layer-wise sensitivity and describe how to integrate these insights into our quantization method.

### 3.1 COLUMN-WISE SENSITIVITY

Column-wise sensitivity refers to the variation in quantization error across columns. This sensitivity is largely driven by systematic outliers (Dettmers et al., 2022; Lee et al., 2024)—values that consistently deviate from the bulk distribution across samples—which distort the shared scale within a quantization block. In microscaling formats like MXINT, where blocks of 32 values share a power-of-two scale, such distortion leads to underutilized dynamic range and higher quantization error. Consequently, columns exhibiting these characteristics require higher bitwidths to preserve overall model accuracy.

We observe that while outliers are the primary cause of column-wise sensitivity, their varying magnitudes (Yi et al., 2025; Lin et al., 2024a) significantly influence the degree of this sensitivity. Prior work has shown that within linear layers, a small subset of highly sensitive columns accounts for a disproportionate share of the overall quantization error (Dettmers et al., 2022; Lee et al., 2024), underscoring the importance of column-level granularity. Specifically, columns containing high-magnitude outliers induce larger errors, as accurately representing such values demands a broader dynamic range. In contrast, columns with milder outliers can be quantized more aggressively without a major loss in accuracy. This variation in outlier magnitudes thus calls for a tailored, column-specific method for bitwidth allocation. By quantifying and exploiting these differences, we can achieve more precise control over quantization.

---

**Algorithm 1** Column-Wise Sensitivity Extraction

---

**Input:** Calibration dataset $\mathcal{D}_{calib}$, Model $\mathcal{M}$
**Output:** Column-wise sensitivities $S_{col}$ for all layers in $\mathcal{M}$
1: $S_{col} \leftarrow \emptyset$          ▷ Initialize column-wise sensitivities
2: **for all** linear layer $L_i \in \mathcal{M}$ **do**
3:     $d_{in} \leftarrow$ input_dim$(L_i)$
4:     $H_i \leftarrow \mathbf{0}^{d_{in} \times d_{in}}$          ▷ Initialize Hessian matrix for layer $L_i$
5:     **for all** sample in $\mathcal{D}_{calib}$ **do**
6:        $X_i \leftarrow$ get_activations$(L_i, \text{sample})$
7:        $H_i \leftarrow H_i + X_i^T X_i$          ▷ Accumulate Hessian approximation
8:     **end for**
9:     $s_i \leftarrow$ diag$(H_i)$          ▷ Sensitivity is the diagonal of the Hessian
10:    Add $s_i$ to $S_{col}$
11: **end for**
12: **return** $S_{col}$

---

To support targeted bitwidth allocation, we formally define column-wise sensitivity based on the statistical properties of tensor distributions. We begin by analyzing the case where quantization is applied to the weights of a linear layer. Let $W$ be the weight matrix, $\hat{W}$ its quantized version, and $X$ the input activations. The quantization-induced error on output channel $i$ is approximated as (Lee et al., 2024):

$$E_i = \|W_{i,:}X - \hat{W}_{i,:}X\|_2^2 \approx \frac{1}{2}\Delta W_{i,:}^T H \Delta W_{i,:} \tag{1}$$

Here, $H = X^T X$ denotes the empirical Hessian matrix of the loss function with respect to the weights, computed from input activations. This approximation implies that the quantization error is influenced by both the weight difference $\Delta W$ and the activation statistics encoded in $H$.

We define the sensitivity of column $j$ as the diagonal entry $H_{j,j}$, which reflects how sensitive the output is to quantization noise in that particular column. This formulation naturally extends to activation quantization. When both weights and activations are quantized, the same Hessian diagonal $H_{j,j} = \sum_{k=1}^n X_{k,j}^2$ serves as a sensitivity indicator. This is because the presence of an outlier in column $j$ directly increases $H_{j,j}$, amplifying the column's sensitivity. Thus, $H_{j,j}$ provides a unified metric for quantization sensitivity for each column $j$ across both weights and activations. To extract these sensitivity scores in practice, we propose Algorithm 1, which uses a calibration dataset to accumulate activation statistics across all linear layers in the model.

### 3.2 LAYER-WISE SENSITIVITY

Layer-wise sensitivity is another critical factor, as layers exhibit varying sensitivity to quantization under identical bitwidth allocation. While most quantization methods apply uniform bitwidth or make only coarse adjustments across layers, our analysis shows that certain layers are more prone to accuracy degradation under low-precision quantization. This variation in sensitivity motivates a more precise bitwidth allocation strategy that accounts for each layer's impact on accuracy and quantization robustness.

To capture this variability, we quantify layer-wise sensitivity by measuring the $L_2$ error at the model's final output when quantization is applied to a single layer in isolation. Algorithm 3 in Appendix A.3 explains how we extract sensitivites in detail. Formally, a linear layer is considered more sensitive if it produces a larger error in the model's final output when quantized under the same bitwidth configuration. Quantization error propagation depends on a layer's position in the network. As shown in Figure 1, quantization errors introduced in earlier layers of the LLaMA-2-7B model (Meta, 2024) compound as they propagate through the

network, amplifying their effect on the final output. In contrast, errors in later layers tend to remain more localized. This position-dependent amplification leads to systematically higher sensitivity in early layers across all linear layer types (Same holds for LLaMA-3-8B, and Mistral-7B-v0.3, while middle layers are more sensitive for Qwen1.5-7B as shown in Appendix A.8). These findings support a sensitivity-aware approach to mixed-precision quantization, where bitwidths are assigned in proportion to a layer's quantization impact.

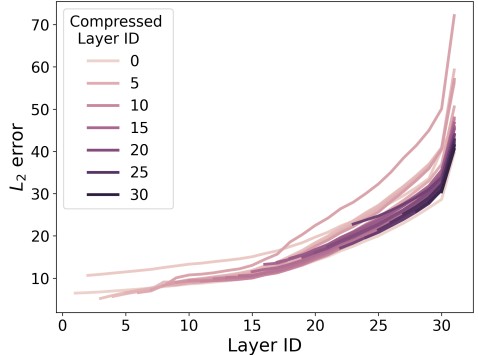

Figure 1: MXINT4 error propagation in q-proj for single-layer quantization on LLaMA-2-7B.

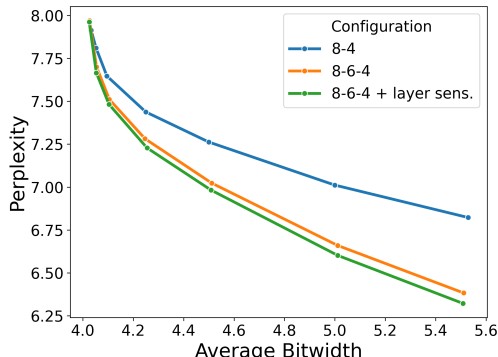

Figure 2: Perplexity dynamics of bitwidth allocation strategies on Llama-3-8B.

## 4 MXSENS: SENSITIVITY-GUIDED TRIPLET QUANTIZATION

In this section, we introduce *MXSens*, a novel quantization method that selectively assigns bitwidths to columns and layers based on their sensitivity to quantization. Unlike prior methods that apply uniform or layer-wise fixed precision, MXSens jointly leverages column- and layer-wise sensitivity to allocate three precision levels—4, 6, and 8 bits—optimally under a given average bitwidth budget. The remainder of this section outlines our method step by step, detailing the motivation behind each design choice.

Quantizing all layers and columns uniformly overlooks the substantial variation in quantization sensitivity, as shown in Section 3. This sensitivity stems from the presence of both *extreme* and *mild outliers*: Extreme outliers are rare but high-magnitude values (Dettmers et al., 2022; Lin et al., 2024a; Yi et al., 2025) that dominate the dynamic range within a block and require high precision, e.g., 8 bits, to avoid severe quantization error. Mild outliers are more frequent and have smaller magnitudes, but they still cause non-negligible degradation when quantized to low bitwidths like MXINT4. Assigning 8-bit precision to the most sensitive columns, with extreme outliers, significantly improves accuracy. However, extending 8-bit allocation to additional columns yields diminishing returns, as only a small fraction of columns actually contain such extreme values. A broader subset of columns, exhibiting moderate sensitivity due to mild outliers, benefits more from intermediate precision, such as 6 bits, enabling more efficient use of the overall bitwidth.

### 4.1 TRIPLET QUANTIZATION

To address the full spectrum of sensitivity across columns, we propose triplet quantization, which assigns three precision levels—4, 6, 8 bits—based on quantization sensitivity. We observe that in each layer, the number of columns containing extreme outliers is typically smaller than the block size of MXINT (32). This makes MXINT particularly well-suited for a mixed-mantissa strategy: we assign 8-bit precision to the top 32

most sensitive columns in each layer, ensuring that all extreme outliers are captured within one high-precision block per layer. This approach aligns with MXINT's block-wise constraints, where each block must use a single mantissa bitwidth to ensure full utilization and efficient hardware execution. Mild outliers are assigned 6-bit mantissas, again in full blocks, while the rest default to 4 bits.

Figure 2 shows that triplet quantization (4-6-8) consistently outperforms the two-precision method (4-8) across a wide range of average bitwidths. The blue curve corresponds to a baseline scheme where the 32 most sensitive columns per layer are assigned 8-bit precision, and all remaining columns default to 4 bits. In contrast, the orange curve illustrates our proposed strategy: the top 32 columns are assigned 8 bits, while the remaining columns are divided between 6 and 4 bits based on column sensitivity and average bitwidth constraints. This finer-grained allocation improves perplexity at similar compression levels, highlighting the effectiveness of including an intermediate bitwidth for mild outliers. Moreover, as shown in Appendix A.4, columns with extreme outliers consistently require 8-bit precision to prevent significant accuracy loss, reinforcing the need for dedicated high-precision blocks in each layer.

## 4.2 COMBINING COLUMN- AND LAYER-WISE SENSITIVITY

While triplet quantization effectively captures column-wise sensitivity by introducing an intermediate 6-bit level, it does not account for global variation across layers. In practice, layers exhibit a wide range of sensitivity to quantization, as shown in Section 3.2. To achieve a globally balanced bitwidth allocation that respects both local (column) and global (layer) sensitivity, we introduce the MXSens method. MXSens extends triplet quantization by jointly considering both column-wise and layer-wise sensitivity to assign bitwidths under a target average precision budget. The method retains the use of 8-bit mantissas for the 32 most sensitive columns in each layer, capturing all extreme outliers while aligning with MXINT's block size. For the remaining columns, MXSens distributes 6- and 4-bit precision in a layer-aware fashion: layers with higher overall sensitivity are assigned more 6-bit columns, while less sensitive layers receive fewer.

---

**Algorithm 2** Sensitivity-Based Bitwidth Allocation

**Input:** Set of column sensitivities $S_{col} = \{s_1, s_2, \dots\}$, layer sensitivity vector $S_{layer}$, average bit-width $B$.
**Output:** Map of column indices $\mathcal{B}_{indices}$ for each bitwidth per layer.

1: $\mathcal{B}_{indices} \leftarrow \emptyset$
2: $d_{in}^{total} \leftarrow \sum_i^n \text{num\_columns}(L_i)$      ▷ Calculate total number of columns in all linear layers
3: $R \leftarrow \frac{(B-4)\cdot\sum_{i=1}^n d_{in}^i - 4\cdot 32\cdot n}{\sum_{i=1}^n 2\cdot S_{layer}^i\cdot(d_{in}^i-32)}$      ▷ Calculate the control parameter for average bitwidth is $B$
4: **for all** linear layer $L_i \in \mathcal{M}$ **do**
5:      $n_i \leftarrow \text{num\_columns}(L_i)$
6:      $I \leftarrow \text{argsort}(S_{col}[i], \text{descending=True})$      ▷ Get column indices sorted by sensitivity
7:      $I_{8bit} \leftarrow I[0:32]$      ▷ Assign 8 bits to top 32 sensitive columns
8:      $N_{6bit} \leftarrow \lfloor R \cdot S_{layer}[i] \cdot (d_{in}^i - 32) \rfloor$      ▷ Calculate number of 6-bit columns
9:      $I_{6bit} \leftarrow I[32 : 32 + N_{6bit}]$      ▷ Assign 6 bits to next $N_{6bit}$ columns
10:      $I_{4bit} \leftarrow I[32 + N_{6bit} :]$      ▷ Assign 4 bits to the rest
11:      $\mathcal{B}_{indices}[i] \leftarrow (I_{8bit}, I_{6bit}, I_{4bit})$      ▷ Store the index lists for the current layer
12: **end for**
13: **return** $\mathcal{B}_{indices}$

---

Algorithm 2 summarizes this strategy. For each layer, columns are first sorted by sensitivity. The top 32 are assigned 8-bit mantissas. Next, the number of 6-bit columns is computed in proportion to the layer's sensitivity score, scaled by a global factor $R$ to meet the target average bitwidth $B$. The remaining columns are assigned 4-bit mantissas. This approach yields a globally optimized bitwidth allocation that aligns precision

with the sensitivity structure of the model, improving both accuracy and hardware overhead. Theorem A.1 in Appendix A.3 shows how $R$ is calculated in a way to ensure the average bitwidth constraint $B$ is satisfied.

## 5 EXPERIMENTAL RESULTS

**Experimental Setup.** We evaluate MXSens across a large range of LLMs and datasets. Our experiments cover models from the LLaMA family (Touvron et al., 2023) (LLaMA-2-7B, LLaMA-2-13B, LLaMA-2-70B, LLaMA-3-8B, LLaMA-3.1-8B), as well as Qwen1.5-7B (Yang et al., 2024), Mistral-7B (Jiang et al., 2023) and Gemma-2-9B (Mesnard et al., 2024). For language modeling tasks, we report perplexities (PPL) on WikiText-2 (Merity et al., 2016); for commonsense reasoning tasks, we report 0-shot accuracies on: Arc-Easy (Clark et al., 2018), Arc-Challenge (Clark et al., 2018), Winogrande (Keisuke et al., 2019), PiQA (Bisk et al., 2020), BoolQ (Clark et al., 2019), HellaSwag (Zellers et al., 2019), OpenBookQA (Mihaylov et al., 2018), and MMLU (Hendrycks et al., 2021). For sensitivity analysis, we use 128 randomly sampled examples from the C4 dataset (Raffel et al., 2019). All experiments are conducted without any retraining or fine-tuning. We compare MXSens with SOTA post-training quantization methods under matched average bitwidths: Two of them are rotation-based, Quarot (Ashkboos et al., 2024b) and RRS (Yi et al., 2025), and two of them are mixed-precision methods, Atom (Zhao et al., 2024) and QUIK (Ashkboos et al., 2024a). Unless otherwise specified, we use the MXINT format and apply quantization to weights, activations, and key-value (KV) caches. In particular, we enforce average bitwidth targets of 4.02-4.03 using the 4-6-8 MXSens configuration, and evaluate all methods under both W4A4KV16 and W4A4KV4 constraints. We provide further details about our experimental setup in Appendix A.2.

### 5.1 MXSENS RESULTS

We evaluate MXSens across a diverse set of models and tasks, and compare its performance to SOTA quantization baselines at matched average bitwidths. For MXSens, we account for the 8-bit exponent overhead shared across 32-element blocks by adding 0.25 bits per element to the average bitwidth. As shown in Tables 1 and 2, MXSens consistently outperforms rotation-based methods (QuaRot, RRS) and mixed-precision strategies (Atom, QUIK), achieving higher 0-shot accuracy and lower PPL across a range of quantization settings and model scales. The reported baseline numbers are taken from their respective papers.

Table 1: 0-shot accuracy (%) on the Common Sense QA tasks and WikiText-2 perplexity.

| Model | Method | Avg. Bits | PQ | ARC-e | ARC-c | HS | WG | Avg. | PPL |
|---|---|---|---|---|---|---|---|---|---|
| LLaMA-2-13B | QuaRot | 4.00 | 78.9 | 72.9 | 46.6 | 76.4 | 70.2 | 69.0 | 5.40 |
| | Atom | 4.25 | 76.5 | 57.5 | 42.3 | 73.8 | 67.4 | 63.5 | 5.31 |
| | QUIK | 4.50 | 79.2 | 74.9 | 48.0 | 78.4 | 71.9 | 70.5 | 5.28 |
| | MXSens | 4.27 | 78.9 | 77.3 | 47.9 | 77.5 | 69.8 | 70.3 | 5.32 |
| | MXSens | 4.50 | 79.4 | 76.5 | 48.6 | 78.4 | 71.3 | 70.8 | 5.09 |
| LLaMA-2-70B | QuaRot | 4.00 | 82.4 | 80.4 | 56.2 | 81.8 | 76.2 | 75.4 | 3.79 |
| | Atom | 4.25 | 79.9 | 58.3 | 46.1 | 79.1 | 74.3 | 67.5 | 3.73 |
| | QUIK | 4.50 | 81.6 | 79.0 | 56.1 | 81.6 | 76.6 | 75.0 | 3.74 |
| | MXSens | 4.27 | 82.2 | 81.8 | 55.8 | 82.6 | 77.0 | 75.9 | 3.77 |
| | MXSens | 4.50 | 82.0 | 81.1 | 57.6 | 83.0 | 76.5 | 76.0 | 3.58 |

Table 1 shows that MXSens consistently outperforms all baselines on both LLaMA-2-13B and LLaMA-2-70B, across accuracy and perplexity (PPL) metrics, under matched average bitwidths. In all configurations, weights, activations, and KV cache are quantized under the same bitwidth budget. On LLaMA-2-13B at ∼4.25 bits, MXSens matches Atom in WikiText-2 PPL (5.32 vs. 5.31) but achieves significantly higher 0-shot accuracy on Common Sense QA tasks (70.3% vs. 63.5%). While QUIK operates at a higher average bitwidth (4.5 bits),

Table 2: 0-shot accuracy (%) on the Common Sense QA tasks.

| #Bits | Model | Method | OBQA | BoolQ | ARC-E | ARC-C | Avg. | PPL |
|---|---|---|---|---|---|---|---|---|
| W4A4KV16 | LLaMA-3-8B | QuaRot | 39.2 | 70.7 | 68.6 | 41.5 | 55.0 | 8.38 |
| | | RRS | **42.4** | 73.6 | 71.1 | 44.8 | 58.0 | 8.11 |
| | | **MXSens** | 42.2 | **77.0** | **75.7** | **47.6** | **60.6** | **7.63** |
| | Mistral | QuaRot | **44.4** | 83.1 | 71.9 | 54.4 | 63.4 | 6.38 |
| | | RRS | 43.4 | **85.1** | 74.4 | **55.4** | **64.6** | 6.31 |
| | | **MXSens** | 42.2 | 81.5 | **78.2** | 49.2 | 62.8 | **5.77** |
| | Qwen1.5-7B | QuaRot | 39.0 | 73.9 | 61.3 | 40.4 | 53.7 | 9.34 |
| | | RRS | **43.0** | 77.7 | 61.5 | 42.0 | 56.0 | **9.17** |
| | | **MXSens** | 40.8 | **78.9** | **67.8** | **42.6** | **57.5** | 9.73 |
| W4A4KV4 | LLaMA-3-8B | QuaRot | 38.6 | 70.6 | 66.7 | 40.4 | 54.1 | 8.76 |
| | | RRS | **42.4** | 73.5 | 68.7 | **44.7** | **57.3** | 8.42 |
| | | **MXSens** | 42.2 | **74.5** | **69.7** | 42.7 | **57.3** | **7.99** |
| | Mistral | QuaRot | 43.2 | 83.0 | 72.8 | 52.5 | 62.9 | 6.45 |
| | | RRS | **45.6** | **84.3** | 73.2 | **54.6** | **64.4** | 6.35 |
| | | **MXSens** | 42.6 | 81.8 | **77.9** | 49.8 | 63.0 | **6.08** |
| | Qwen1.5-7B | QuaRot | 39.6 | 74.3 | 62.1 | 42.0 | 54.5 | 9.55 |
| | | RRS | 40.4 | 76.6 | 61.8 | **42.5** | 55.3 | **9.37** |
| | | **MXSens** | **40.6** | **78.0** | **65.4** | 42.3 | **56.6** | 9.95 |

Table 3: Ablation study: the effect of various average bitwidth configurations for Llama2-7B.

| Precisions | Avg BW | MMLU | A-c | A-e | WG | PQ | BQ | HS | Avg Acc | PPL |
|---|---|---|---|---|---|---|---|---|---|---|
| MXINT3 | 3.28 | 26.9 | 34.7 | 55.6 | 56.4 | 46.0 | 70.9 | 62.6 | 55.5 | 9.37 |
| + | 3.35 | 27.2 | 35.4 | 61.0 | 60.1 | 47.6 | 71.5 | 63.9 | 57.3 | 8.31 |
| MXINT6 | 3.50 | 27.0 | 36.8 | 62.0 | 62.0 | 48.0 | 72.2 | 65.9 | 58.8 | 7.76 |
| + | 3.75 | 28.3 | 37.7 | 63.8 | 62.5 | 50.0 | 73.2 | 67.3 | 60.1 | 7.30 |
| MXINT8 | 4.00 | 28.7 | 39.7 | 66.1 | 64.2 | 50.6 | 73.7 | 69.0 | 60.5 | 6.91 |
| MXINT4 | 4.27 | 36.7 | 43.3 | 70.7 | 66.5 | 76.8 | 73.9 | 74.1 | 67.6 | 5.98 |
| + | 4.35 | 37.3 | 44.5 | 72.1 | 68.3 | 76.8 | 74.4 | 73.7 | 68.3 | 5.90 |
| MXINT6 | 4.75 | 38.7 | 44.2 | 72.4 | 66.8 | 77.1 | 75.0 | 74.1 | 68.3 | 5.76 |
| + | 5.25 | 40.3 | 45.3 | 73.0 | 68.4 | 77.0 | 74.9 | 75.0 | 68.9 | 5.64 |
| MXINT8 | 5.75 | 40.8 | 45.3 | 73.5 | 69.3 | 77.6 | 77.0 | 75.5 | 69.7 | 5.49 |
| FP16 | 16.00 | 41.9 | 46.2 | 74.5 | 69.3 | 78.1 | 77.8 | 76.0 | 70.3 | 5.42 |

MXSens not only achieves comparable accuracy at 4.27 bits, but outperforms QUIK in both accuracy and PPL when evaluated at 4.5 bits, demonstrating better compression-efficiency trade-offs. While QuaRot achieves 69.0% accuracy on LLaMA-2-13B, it is the closest competitor to MXSens on LLaMA-2-70B; however, MXSens still surpasses it in both accuracy (77.0% vs. 76.2%) and PPL at 4.28 bits.

Table 2 reports zero-shot accuracy for three models (LLaMA-3-8B, LLaMA-3.1-8B, and Mistral) on four commonsense reasoning benchmarks: OpenBookQA (OBQA), BoolQ, ARC-Easy, and ARC-Challenge; and perplexity on WikiText-2. MXSens outperforms RRS and QuaRot on the majority of tasks, demonstrating strong generalization beyond language modeling. Similar to its perplexity performance, MXSens enables LLaMA-3-8B to achieve higher accuracy compared to both baselines across all evaluated tasks. For Mistral-7B, MXSens underperforms QuaRot and RRS on three tasks, but surpasses both baselines on ARC-E accuracy and WikiText-2 perplexity. We report additional results on other models and benchmarks in the Appendix.

## 5.2 ABLATION STUDY ON THE AVERAGE NUMBER OF BITS

Table 3 examines how LLaMA-2-7B's zero-shot accuracy on seven diverse benchmarks and its WikiText-2 perplexity change as we vary the average bitwidth under two mixed-precision templates, "3-6-8" and "4-6-8." As the average bitwidth increases from roughly 3.28 to 4 bits in the 3-6-8 configuration, overall accuracy climbs steadily from 55.5% to 60.5%. The higher-floor 4-6-8 template exhibits a similar trend. These gains show that each additional bit can contribute meaningfully to model quality across different bit-budget regimes. At 5.75 bits, the accuracy gap narrows, achieving 69.7% accuracy, only 0.6 points below FP16, showing that sensitivity-guided mixed precision can almost fully match full-precision quality at just one-third of the bit cost. Together, these results demonstrate MXSens's flexibility across a wide range of bitwidth budgets.

## 6 HARDWARE IMPLICATIONS

Our method quantizes weights, activations, and KV cache using the MXINT format (Drumond et al., 2018; Rouhani et al., 2023b), which is designed for dense execution in hardware. Emerging commercial GPUs and neural processing units (NPUs) such as NVIDIA Blackwell and Qualcomm AI100 have native support for microscaling formats. Prior work (Drumond et al., 2021; Gil et al., 2025) has also demonstrated that hardware accelerators with native support for microscaling formats can deliver the same training and inference throughput as integer arithmetic.

There are multiple ways to support MXSens' mixed-mantissa computation on commercial GPUs and NPUs. First, several proposals (Zhang et al., 2022; Noh et al., 2023) use custom hardware units to support multiple mantissa bitwidths in the same data-path, enabling no loss in throughput or latency. Second, commercial GPUs typically support multiple precisions of the same data type (e.g., INT4 and INT8 support in tensor cores). Similar to prior work (Dettmers et al., 2022; Zhao et al., 2024; Ramachandran et al., 2025) that decomposes a tensor into high-precision and low-precision tensors and multiplies them concurrently, MXSens can also be implemented similarly with minimal latency or throughput overhead. Third, even on hardware with a fixed-precision 4-bit MXINT datapath, there are proposals (Gil et al., 2025; Harma et al., 2022) to emulate high-precision arithmetic using the low-precision arithmetic units. While each high-precision operation takes $4\times$ more cycles than a low-precision operation, the relatively small number of these operations results in a minimal latency overhead. At a bitwidth budget of 4.03, only 1% of all arithmetic operations require higher precision, resulting in an estimated 3% latency overhead. One of our key contributions is maintaining model performance within a 4.03-bit budget, avoiding the high latency of larger bitwidths.

MXSens does increase memory usage, but enforcing a bitwidth budget of 4.03 bits limits the increase to less than 1%. Overall, implementing MXSens in commercial GPUs and accelerators is relatively straightforward. The performance impact of using variable-bitwidth activations is marginal, while enabling significant accuracy gains. Our design leverages the fact that most activations and weights can be quantized to four bits, with varying levels of higher precision reserved for columns identified as sensitive–achieving an optimal trade-off between accuracy and hardware efficiency.

## 7 CONCLUSION

We introduce MXSens, a sensitivity-guided, mixed-precision quantization framework for LLMs that delivers state-of-the-art performance without requiring architectural changes or retraining. By leveraging Hessian-guided fine-grained sensitivity analysis across both layers and columns, MXSens effectively allocates 8-, 6-, and 4-bit precision to balance compression and accuracy. Our extensive experiments show that MXSens consistently outperforms existing methods on both language modeling and commonsense reasoning tasks across a wide range of models. Our results underscore the value of sensitivity-aware quantization in pushing the boundaries of efficient LLM inference.

## 8 REPRODUCIBILITY STATEMENT

We have made extensive efforts to ensure the reproducibility of our results. Experimental setup details, including models, datasets, and evaluation metrics; as well as all implementation details of MXSens, including sensitivity computation, bitwidth allocation, and MXINT quantization routines, are provided in Section 5 and Appendix A.2. Pseudocodes for the column-wise sensitivity extraction, layer-wise sensitivity extraction, and triplet quantization algorithm are included in Algorithms 1, 3, 2, respectively. Hardware assumptions for MXINT execution are discussed in Section 6. We also provide an anonymous code repository containing all scripts required to reproduce our main results, which is available as part of the supplementary materials.

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

## A APPENDIX

### A.1 LANGUAGE MODEL USAGE IN THE PAPER

Language models were used to improve writing clarity, correct grammar, and typos, and ensure compliance with the ICLR author guidelines. Apart from their use in benchmark evaluations, language models were not involved in any other part of this work.

### A.2 EXPERIMENTAL DETAILS

We use the `lm-evaluation-harness` framework (Gao et al., 2024) to evaluate commonsense reasoning tasks and adapt the perplexity evaluation code from the QuaRot repository (Ashkboos et al., 2024b). Following prior work (Yi et al., 2025; Zhao et al., 2024), we report `acc_norm` for tasks where it is available, and standard `acc` otherwise. We emulate the MXINT numerical format using PyTorch (Paszke et al., 2017) and model implementations are based on the `transformers` library (Wolf et al., 2020).

All experiments are conducted on a single NVIDIA A100 GPU (Nvidia, 2020), except for LLaMA-2-70B, which requires multiple GPUs due to its memory demands. We quantize the weights and activations in all linear layers within the decoder, including both MLP and attention layers. For KV-cache quantization, we apply a 4-8 bitwidth configuration with an average bitwidth fixed at 4.03, as these components are not significantly affected by mild outliers. All remaining layers, including Softmax, LayerNorm, embedding layers, and the `lm_head` layer, are kept in FP16 format.

### A.3 FORMAL DEFINITION OF BITWIDTH ALLOCATION

---

**Algorithm 3** Layer-Wise Sensitivity Extraction

---

**Input:** Calibration dataset $\mathcal{D}_{calib}$, Model $\mathcal{M}$
**Output:** Normalized layer-wise sensitivities $S_{layer}$ for all layers in $\mathcal{M}$
1: $E \leftarrow []$                 ▷ Initialize list of errors
2: $X_{fp32} \leftarrow \mathcal{M}(\mathcal{D}_{calib})$        ▷ Get final activations from original FP32 model
3: **for all** linear layer $L_i \in \mathcal{M}$ **do**
4:    $\widetilde{\mathcal{M}_i} \leftarrow$ quantize_layer$(\mathcal{M}, L_i)$         ▷ Quantize only layer $L_i$
5:    $\widetilde{X_i} \leftarrow \widetilde{\mathcal{M}_i}(\mathcal{D}_{calib})$      ▷ Get final activations from modified model
6:    $e_i \leftarrow \|X_{fp32} - \widetilde{X_i}\|_2^2$        ▷ Compute L2 error at the output
7:    $E \leftarrow E \cup [e_i]$
8: **end for**
9: $e_{max} \leftarrow \max(E)$         ▷ Find maximum error for normalization
10: $S_{layer} \leftarrow [e_i/e_{max}$ for $e_i \in E]$       ▷ Scale errors to a $[0, 1]$ range
11: **return** $S_{layer}$

---

**Theorem A.1.** *Let $B$ be the average bitwidth, $n$ - number of layers, $d_{in}^i$ - input dimention of the $i$-th linear layer, $S_{layer}^i \in [0, 1]$ - sensitivity of the $i$-th layer. Consider a scenario of triplet allocation with 8, 6 and 4 bits, where 8 bits are allocated to the top 32 sensitive columns, and 6 and 4 bits are allocated to the rest. Let $R$ be the allocation control parameter that affects the ratio between the number of 6-bit columns and 4-bit columns in the layer. Then, to achieve the average bitwidth $B$, the allocation control parameter can be calculated using the following formula:*

$$R = \frac{(B - 4) \cdot \sum_{i=1}^n d_{in}^i - 4 \cdot 32 \cdot n}{\sum_{i=1}^n 2 \cdot S_{layer}^i \cdot (d_{in}^i - 32)} \tag{2}$$

*Proof.* Average bitwidth $B$ can be calculated using the following formula:

$$B = \frac{\sum_{i=1}^{n}(8 \cdot \overbrace{32}^{\text{\# of 8-bit}} + 6 \cdot \overbrace{(R \cdot S_{layer}^i) \cdot (d_{in}^i - 32)}^{\text{\# of 6-bit}} + 4 \cdot \overbrace{(1 - R \cdot S_{layer}^i) \cdot (d_{in}^i - 32))}^{\text{\# of 4-bit}}}{\sum_{i=1}^{n} d_{in}^i} \tag{3}$$

$$= \frac{8 \cdot 32 \cdot n + \sum_{i=1}^{n}(2 \cdot R \cdot S_{layer}^i + 4) \cdot (d_{in}^i - 32)}{\sum_{i=1}^{n} d_{in}^i} \tag{4}$$

From this formula, we can deduce the control parameter $R$:

$$\frac{8 \cdot 32 \cdot n + \sum_{i=1}^{n}(2 \cdot R \cdot S_{layer}^i + 4) \cdot (d_{in}^i - 32)}{\sum_{i=1}^{n} d_{in}^i} = B \tag{5}$$

$$8 \cdot 32 \cdot n + \sum_{i=1}^{n}(2 \cdot R \cdot S_{layer}^i + 4) \cdot (d_{in}^i - 32) = B \cdot \sum_{i=1}^{n} d_{in}^i \tag{6}$$

$$\sum_{i=1}^{n}(2 \cdot R \cdot S_{layer}^i + 4) \cdot (d_{in}^i - 32) = B \cdot \sum_{i=1}^{n} d_{in}^i - 8 \cdot 32 \cdot n \tag{7}$$

$$R \cdot \sum_{i=1}^{n} 2 \cdot S_{layer}^i \cdot (d_{in}^i - 32) + \sum_{i=1}^{n} 4 \cdot (d_{in}^i - 32) = B \cdot \sum_{i=1}^{n} d_{in}^i - 8 \cdot 32 \cdot n \tag{8}$$

$$R \cdot \sum_{i=1}^{n} 2 \cdot S_{layer}^i \cdot (d_{in}^i - 32) = B \cdot \sum_{i=1}^{n} d_{in}^i - 8 \cdot 32 \cdot n - \sum_{i=1}^{n} 4 \cdot (d_{in}^i - 32) \tag{9}$$

$$R \cdot \sum_{i=1}^{n} 2 \cdot S_{layer}^i \cdot (d_{in}^i - 32) = (B - 4) \cdot \sum_{i=1}^{n} d_{in}^i - 4 \cdot 32 \cdot n \tag{10}$$

$$R = \frac{(B - 4) \cdot \sum_{i=1}^{n} d_{in}^i - 4 \cdot 32 \cdot n}{\sum_{i=1}^{n} 2 \cdot S_{layer}^i \cdot (d_{in}^i - 32)} \tag{11}$$

$\square$

### A.4 OUTLIERS NEED 8 BITS

Columns containing systematic outliers consistently require 8-bit precision to avoid significant accuracy degradation, regardless of the layer in which they appear. To validate this hypothesis, we assign 6-bit precision to the 32 most sensitive columns in each linear layer of Llama-2-7B, as determined by their sensitivity scores, and quantize all remaining columns to 4 bits. We choose 32 columns to match the MXINT block size, as allocating fewer would underutilize the block without reducing overhead. Next, we incrementally increase the precision of these sensitive columns from 6 bits to 8 bits, beginning with the most sensitive layers and tracking the change in model perplexity on the WikiText-2 dataset.

Figure 3 illustrates these results. Each dot represents a configuration with a different subset of layers using 8-bit precision, while others remain at 6 bits. The leftmost point corresponds to all layers using only 6 and 4 bits, while the rightmost point represents the configuration where all eligible layers use 8 and 4 bits. We observe that enabling 8-bit precision for the most sensitive columns leads to a substantial reduction in PPL error. Although the curve flattens as more layers adopt 8-bit precision, the PPL error still decreases notably–from 0.5 to 0.3–with only a marginal increase in average bitwidth (approximately 0.005 bits). Given this favorable trade-off, we store the top 32 most sensitive columns in every layer using 8-bit precision.

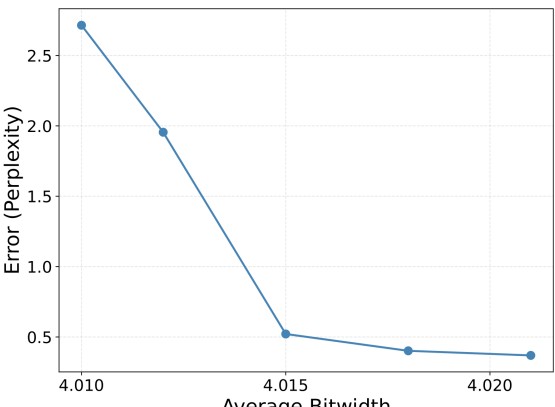

Figure 3: Perplexity error dynamics when upgrading outlier columns from 6 to 8 bits across linear layers.

## A.5 ADDITIONAL MXSENS RESULTS

In this section, we provide additional results for MXSens on eight commonsense reasoning tasks: MMLU, ARC-Challenge, ARC-Easy, WinoGrande, PiQA, BoolQ, HellaSwag, and OpenBookQA. Table 4 reports the 0-shot accuracy for five models–LLaMA-2-13B, LLaMA-2-70B, LLaMA-3-8B, Mistral-7B, and Qwen1.5-7B–under both W4A4KV4 and W4A4KV16 configurations. These results further validate the generalization capabilities of MXSens across architectures and bitwidth settings.

Table 4: 0-shot accuracy (%) on various Common Sense QA tasks using MXSens.

| #Bits | Model | MMLU | A-c | A-e | WG | PQ | BQ | HS | OBQA |
|---|---|---|---|---|---|---|---|---|---|
| W4A4KV16 | LLaMA-2-13B | 48.5 | 48.7 | 77.9 | 70.0 | 79.9 | 79.8 | 77.6 | 44.2 |
| | LLaMA-2-70B | 63.1 | 56.3 | 81.9 | 77.2 | 81.8 | 82.3 | 82.7 | 47.4 |
| | LLaMA-3-8B | 54.8 | 47.6 | 75.7 | 69.6 | 77.1 | 77.0 | 75.4 | 42.2 |
| | Mistral | 54.0 | 49.1 | 78.2 | 69.5 | 80.1 | 81.5 | 78.5 | 42.2 |
| | Qwen1.5-7B | 55.1 | 42.6 | 67.8 | 62.9 | 74.6 | 78.9 | 72.8 | 40.8 |
| W4A4KV4 | LLaMA-2-13B | 48.7 | 47.9 | 77.3 | 69.8 | 78.9 | 80.0 | 77.5 | 43.0 |
| | LLaMA-2-70B | 63.3 | 55.8 | 81.8 | 77.0 | 82.2 | 82.0 | 82.6 | 48.2 |
| | LLaMA-3-8B | 51.6 | 42.7 | 69.7 | 67.3 | 72.9 | 74.5 | 74.4 | 42.2 |
| | Mistral | 51.9 | 49.8 | 77.9 | 69.5 | 78.6 | 81.8 | 78.5 | 42.6 |
| | Qwen1.5-7B | 53.9 | 42.3 | 65.4 | 61.1 | 74.4 | 78.0 | 72.3 | 40.6 |

## A.6 MXSENS WIKITEXT-2 RESULTS

We evaluate MXSens on the WikiText-2 dataset under two standard quantization settings: W4A4KV16 and W4A4KV4. Table 5 reports perplexity results across a diverse set of models at a fixed average bitwidth of approximately 4.02. MXSens consistently outperforms all baselines, including RRS and QuaRot, demonstrating the effectiveness of sensitivity-guided bitwidth allocation. In the W4A4KV4 setting, MXSens achieves perplexities of 7.63 and 7.65 on LLaMA-3-8B and LLaMA-3.1-8B, respectively—substantially improving over RRS (8.11 and 8.12) and QuaRot (8.38 on both models). The gains are even more pronounced in

the W4A4KV16 setting, where MXSens achieves perplexities of 7.63 and 7.65. While MXSens exhibits marginally higher perplexity than RRS and QuaRot on Qwen-1.5-7B, this does not translate to reduced performance on zero-shot tasks.

Table 5: Comparison of MXSens against state-of-the-art quantization methods on WikiText-2 perplexity of various models. We evaluate models and methods on two quantization schemes: A4W4KV4 and A4W4KV16.

| #Bits W-A-KV | Method | LLaMA 2-13B | LLaMA 2-70B | LLaMA 3-8B | LLaMA 3.1-8B | Mistral 7B | Qwen 1.5-7B |
|---|---|---|---|---|---|---|---|
| W16A16KV16 | FP16 | 4.88 | 3.32 | 6.13 | 6.24 | 5.25 | 7.95 |
| | QuaRot | 5.39 | 3.85 | 8.38 | 8.38 | 6.38 | 9.34 |
| W4A4KV16 | RRS | 5.36 | 3.86 | 8.11 | 8.12 | 6.31 | **9.17** |
| | MXSens | **5.24** | **3.72** | **7.63** | **7.65** | **5.77** | 9.73 |
| | QuaRot | 5.51 | 3.89 | 8.76 | 8.80 | 6.45 | 9.55 |
| W4A4KV4 | RRS | 5.45 | 3.89 | 8.42 | 8.49 | 6.35 | **9.37** |
| | MXSens | **5.32** | **3.77** | **7.99** | **8.07** | **6.08** | 9.95 |

## A.7 ALLOCATION OF 8-BIT COLUMNS

As previously discussed, we fix the number of 8-bit columns to 32, while the number of 6-bit and 4-bit columns varies according to layer sensitivity. To validate that 32 is the optimal number of 8-bit columns for our allocation scheme, we conduct an ablation study. We evaluate the model's performance using MXSens with the number of 8-bit columns ranging from 32 to 160 in increments of 32. Throughout this experiment, the MXINT block size is held constant at 32 elements. The results are illustrated in Figure 4. The figure shows that an allocation strategy with 32 8-bit columns is Pareto-efficient, achieving maximum model performance at the lowest average bitwidth. Although allocating more 8-bit columns increases the average bitwidth, it does not yield any significant improvement in model performance. Therefore, we conclude that using 32 8-bit columns is optimal for our method.

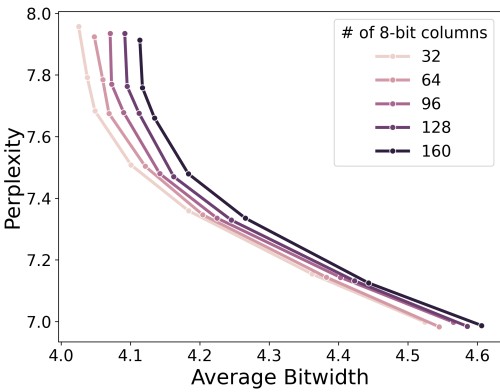

Figure 4: Perplexity error dynamics when changing the number of 8-bit columns.

## A.8 LAYER SENSITIVITY ERROR DYNAMICS

Figures 5, 6 and 7 shows error dynamics for LLaMA-3-8B, Mistral-7B-v0.3, and Qwen1.5-7B as mentioned in Section 3.2.

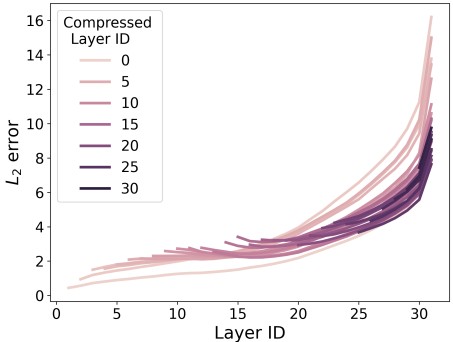 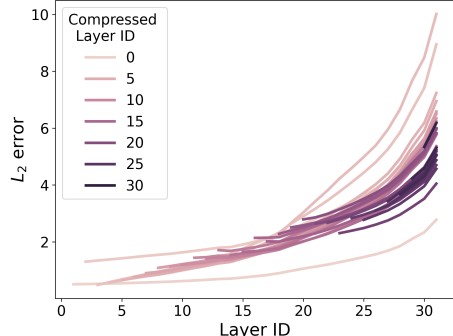

Figure 5: Error dynamics for LLaMA-3-8B     Figure 6: Error dynamics for Mistral-7B-v0.3

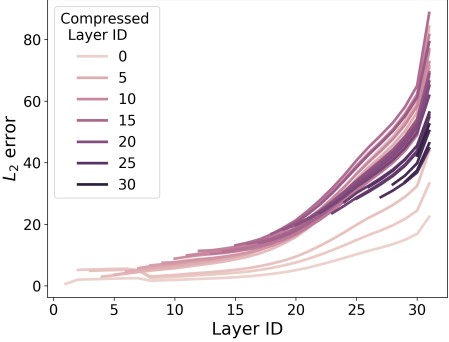

Figure 7: Error dynamics for Qwen1.5-7B

## A.9 CALIBRATION DATA

Table 6 shows the results of our ablation study on the choice of calibration dataset. We compare the effects of using Wikitext-2 and C4 as a calibration dataset. We observe that both datasets produce similar results in terms of average accuracy and perplexity.

## A.10 ADDITIONAL RESULTS FOR THE ABLATION STUDY

In the main text, we evaluate the impact of average bitwidth on LLaMA-2-7B's performance under two mixed-precision configurations (3-6-8 and 4-6-8), demonstrating how accuracy changes with varying average

Table 6: Ablation study on the calibration dataset. Perplexity scores are reported for different average bitwidths, showing a negligible impact from the choice of calibration dataset (Wikitext-2 vs. C4). Perplexity was measured on WikiText-2 dataset.

| Avg. Bitwidth | Calibration Dataset | A-e | A-c | BQ | OBQA | Avg. Acc. | PPL |
|---|---|---|---|---|---|---|---|
| 4.03 | Wikitext-2 | 69.15 | 45.48 | 75.54 | 42.00 | 58.04 | 7.75 |
| | C4 | 70.58 | 45.22 | 76.18 | 41.00 | 58.25 | 7.85 |

bitwidth. In this appendix, we extend this analysis with additional ablation experiments across a broader set of models. Table 7, 8, 9,and 10 present these results for Qwen-1.5-7B, LLaMA-2-13B, LLaMA-3.2-1B, and LLaMA-3.2-3B respectively. These results provide deeper insight into how fine-grained precision control contributes to model quality under varying bitwidth constraints and across diverse models.

Table 7: Ablation study to study the effect of various average bitwidth configurations. We evaluate 0-shot accuracy (%) on the Common Sense QA tasks and WikiText2 perplexity for Qwen 1.5-7B.

| Precisions | Avg BW | MMLU | A-c | A-e | WG | PQ | BQ | HS | Avg Acc | WT-2 PPL |
|---|---|---|---|---|---|---|---|---|---|---|
| MXINT3 + MXINT6 + MXINT8 | 3.03 | 34.6 | 33.0 | 48.6 | 55.5 | 65.9 | 63.6 | 52.6 | 50.5 | 30.31 |
| | 3.10 | 38.4 | 33.0 | 51.0 | 58.9 | 69.1 | 60.1 | 62.4 | 53.3 | 13.41 |
| | 3.25 | 40.7 | 35.4 | 56.3 | 59.1 | 71.1 | 59.1 | 64.1 | 55.1 | 12.36 |
| | 3.50 | 42.9 | 35.4 | 57.8 | 58.6 | 71.2 | 64.6 | 66.3 | 56.7 | 11.31 |
| | 3.75 | 45.2 | 35.8 | 59.2 | 59.3 | 73.1 | 67.1 | 67.5 | 58.2 | 10.65 |
| MXINT4 + MXINT6 + MXINT8 | 4.03 | 55.2 | 42.5 | 62.3 | 59.6 | 74.8 | 78.8 | 73.0 | 63.7 | 9.73 |
| | 4.10 | 56.0 | 42.0 | 60.4 | 61.9 | 76.7 | 75.5 | 74.4 | 63.8 | 8.77 |
| | 4.50 | 57.4 | 41.6 | 61.3 | 62.2 | 76.4 | 79.9 | 74.6 | 64.8 | 8.49 |
| | 5.00 | 57.8 | 40.8 | 62.2 | 65.2 | 76.8 | 81.8 | 75.7 | 65.8 | 8.25 |
| | 5.50 | 58.5 | 41.6 | 61.9 | 65.2 | 78.0 | 82.5 | 76.3 | 66.3 | 8.09 |
| FP16 | 16.00 | 59.6 | 42.8 | 62.2 | 66.1 | 78.4 | 82.5 | 77.0 | 66.9 | 7.95 |

Table 8: Ablation study to study the effect of various average bitwidth configurations. We evaluate 0-shot accuracy (%) on the Common Sense QA tasks and WikiText2 perplexity for Llama2-13B.

| Precisions | Avg BW | MMLU | A-c | A-e | WG | PQ | BQ | HS | Avg Acc | WT-2 PPL |
|---|---|---|---|---|---|---|---|---|---|---|
| MXINT3 | 3.03 | 24.1 | 27.4 | 45.5 | 64.6 | 51.1 | 73.1 | 51.5 | 60.4 | 7.17 |
| + | 3.10 | 24.5 | 28.5 | 45.6 | 66.1 | 52.3 | 74.2 | 52.5 | 62.2 | 6.67 |
| MXINT6 | 3.25 | 24.7 | 28.9 | 49.7 | 67.3 | 53.2 | 74.3 | 56.1 | 63.1 | 6.39 |
| + | 3.50 | 26.8 | 31.2 | 56.4 | 66.8 | 54.4 | 76.4 | 58.8 | 64.9 | 6.12 |
| MXINT8 | 3.75 | 28.5 | 36.2 | 59.3 | 69.0 | 55.1 | 76.0 | 63.6 | 65.3 | 5.88 |
| MXINT4 | 4.02 | 47.0 | 46.2 | 74.5 | 71.0 | 77.0 | 80.5 | 76.1 | 70.9 | 5.24 |
| + | 4.10 | 47.0 | 48.4 | 75.2 | 70.4 | 78.3 | 80.1 | 76.4 | 71.5 | 5.18 |
| MXINT6 | 4.50 | 48.0 | 46.9 | 75.0 | 70.4 | 78.3 | 80.1 | 77.1 | 71.3 | 5.08 |
| + | 5.00 | 49.5 | 48.6 | 76.1 | 71.5 | 78.2 | 80.3 | 78.0 | 72.1 | 5.00 |
| MXINT8 | 5.50 | 50.6 | 49.4 | 77.4 | 71.6 | 78.5 | 80.7 | 78.5 | 72.7 | 4.93 |
| FP16 | 16.00 | 52.1 | 49.1 | 77.5 | 72.1 | 79.1 | 80.6 | 79.4 | 73.0 | 4.88 |

Table 9: Ablation study to study the effect of various average bitwidth configurations. We evaluate 0-shot accuracy (%) on the Common Sense QA tasks and WikiText2 perplexity for Llama3.2-1B.

| Precisions | Avg BW | MMLU | A-c | A-e | WG | PQ | BQ | HS | Avg Acc | WT-2 PPL |
|---|---|---|---|---|---|---|---|---|---|---|
| MXINT3 | 3.03 | 24.0 | 24.8 | 28.0 | 50.7 | 53.9 | 53.3 | 27.5 | 37.5 | 1097.66 |
| + | 3.10 | 23.4 | 24.7 | 29.8 | 51.3 | 53.3 | 54.6 | 28.4 | 37.9 | 716.09 |
| MXINT6 | 3.25 | 23.8 | 22.0 | 30.7 | 51.5 | 54.0 | 58.0 | 29.1 | 38.4 | 365.80 |
| + | 3.50 | 23.8 | 25.3 | 33.3 | 50.8 | 56.1 | 58.0 | 31.4 | 39.8 | 199.17 |
| MXINT8 | 3.75 | 24.1 | 23.6 | 36.1 | 50.5 | 57.5 | 58.2 | 34.4 | 40.6 | 90.16 |
| MXINT4 | 4.03 | 26.7 | 30.3 | 50.6 | 51.6 | 66.3 | 50.9 | 53.5 | 47.1 | 15.17 |
| + | 4.10 | 26.7 | 31.0 | 52.0 | 54.7 | 67.4 | 57.6 | 54.7 | 49.2 | 14.36 |
| MXINT6 | 4.50 | 29.2 | 34.0 | 53.2 | 56.2 | 69.7 | 60.2 | 57.4 | 51.4 | 12.39 |
| + | 5.00 | 32.9 | 33.0 | 56.6 | 57.3 | 71.9 | 60.4 | 59.8 | 53.1 | 11.11 |
| MXINT8 | 5.50 | 35.3 | 35.2 | 59.8 | 59.5 | 74.5 | 63.5 | 62.0 | 55.7 | 10.32 |
| FP16 | 16.00 | 37.6 | 36.3 | 60.4 | 60.5 | 74.4 | 64.0 | 63.7 | 56.7 | 9.75 |

Table 10: Ablation study to study the effect of various average bitwidth configurations. We evaluate 0-shot accuracy (%) on the Common Sense QA tasks and WikiText2 perplexity for Llama3.2-3B.

| Precisions | Avg BW | MMLU | A-c | A-e | WG | PQ | BQ | HS | Avg Acc | WT-2 PPL |
|---|---|---|---|---|---|---|---|---|---|---|
| MXINT3 | 3.03 | 23.1 | 25.3 | 32.3 | 53.6 | 55.7 | 55.6 | 35.1 | 40.1 | 94.77 |
| + | 3.10 | 24.7 | 25.0 | 38.5 | 51.1 | 59.1 | 52.4 | 38.2 | 41.3 | 53.76 |
| MXINT6 | 3.25 | 25.0 | 26.5 | 41.3 | 53.7 | 61.4 | 49.0 | 44.0 | 43.0 | 31.44 |
| + | 3.50 | 26.9 | 27.9 | 44.9 | 53.6 | 63.8 | 55.2 | 50.6 | 46.1 | 21.12 |
| MXINT8 | 3.75 | 28.2 | 30.5 | 50.1 | 53.0 | 64.5 | 54.7 | 54.3 | 47.9 | 16.68 |
| MXINT4 | 4.03 | 43.1 | 40.3 | 63.5 | 61.2 | 73.2 | 64.7 | 68.1 | 59.2 | 9.95 |
| + | 4.10 | 45.8 | 40.1 | 63.6 | 62.5 | 73.8 | 69.2 | 68.9 | 60.5 | 9.55 |
| MXINT6 | 4.50 | 47.9 | 41.4 | 68.3 | 66.0 | 74.6 | 72.3 | 70.3 | 63.0 | 8.83 |
| + | 5.00 | 50.8 | 42.9 | 68.2 | 67.2 | 75.8 | 72.0 | 71.8 | 64.1 | 8.33 |
| MXINT8 | 5.50 | 52.6 | 45.1 | 69.3 | 68.0 | 76.4 | 72.5 | 73.0 | 65.3 | 8.05 |
| FP16 | 16.00 | 54.4 | 46.1 | 71.7 | 69.8 | 76.7 | 73.2 | 73.7 | 66.5 | 7.81 |

