# OpenReview forum: "MXSens: Sensitivity-Aware Mixed-Precision Quantization for Efficient LLM Inference"
_ICLR.cc/2026/Conference — Submitted to ICLR 2026_

### Official Review · Reviewer_DuvT · 2025-10-20

**Soundness:** 2
**Presentation:** 3
**Contribution:** 1
**Rating:** 2
**Confidence:** 5

**Summary:**

In this work, the authors introduce MXSens, in which they use the diagonal Hessian values for determining the sensitivity of each column, and assign outlier columns with higher precision, such as 8-bit or 6-bit. Essentially, the precision is mixed across layers and also within layers, which makes MXSens a fine-grained approach. However, the reviewer is particularly concerned with its hardware compatibility and novelty.

**Strengths:**

Well-explained approach with clear motivation and algorithmic code.

**Weaknesses:**

My major concern for this work is its novelty.

+ Mixed-precision quantization has been a well-explored topic, with the first work (HAQ) published in 2018. In this work, the authors used a standard bit-search strategy to assign the bitwidth across layers. The outlier column, with higher precision, will gain significant benefit as shown in SpQR. After reading the methodology, one can already expect the experimental results.

+ A different motivation from work seems to be the mixed-precision on microscaling formats. However, the reviewer raised significant concerns about this design choice. Indeed, microscaling formats are supported on hardware, but that does not mean mixed-precision microscaling formats are supported. I can imagine huge kernel design efforts are needed when implementing your formats on Blackwell GPUs. In your W4A4 microscaling formats, the key is that each tile can perform matmul for each scaling. But with mixed-precision, each tile is not synchronous.

+ Hessian diagonal does not seem to be a reasonable metric for your mixed-precision, as the weight quantization error also affects the overall layer output error. Moreover, Slim-LLM has proposed other metrics to allocate mixed-precision for different columns.

+ In current LLM literature, perplexity has become less important. I'd like to see reasoning task accuracy and MMLU results to evaluate the method's effectiveness.

**Questions:**

None

---

### Official Review · Reviewer_SpFs · 2025-10-31

**Soundness:** 2
**Presentation:** 3
**Contribution:** 1
**Rating:** 2
**Confidence:** 5

**Summary:**

This paper studies the mixed-precision quantization for LLMs. The background is that different columns/layers matter differently to the quantization. It uses the per-layer Hessian information to identify the precision of each column and assign different portions of higher-bits to different layers according to their sensitivity to the end-to-end loss. It also explores the utilization of microscaling data format to support effective quantization. It presents the evaluation to demonstrate its efficacy.

**Strengths:**

1. It focuses on a very important topic, the compression of LLMs.
2. The presentation is very clear.

**Weaknesses:**

1. The idea is not new, e.g., the sensitivity identification of the columns within each layer is exact the same with Atom [1]
2. It has missed the discussion and evaluation of some highly related mixed-precision quantization work, e.g., SliM-LLM [2], MixLLM[3]
3. It has overclaimed the contribution, e.g., assigning different portion of bits for different layers is already explored in MixLLM [3]
4. It has extensively discussed the microscaling format, but there is no noval contribution based on this data format. This is just to make use of a data format.

[1] Atom: Low-Bit Quantization for Efficient and Accurate LLM Serving

[2] SliM-LLM: Salience-Driven Mixed-Precision Quantization for Large Language Models

[3] MixLLM: LLM Quantization with Global Mixed-precision between Output-features and Highly-efficient System Design

**Questions:**

The mixed-precision quantization is extensively studied in recent years. This paper does not provide new insights upon the state-of-the-art.
- It could be better to compare this work with the recent state-of-the-art, like SliM-LLM and MixLLM mentioned above.
- Section 3.1 uses the same method with Atom. You can check the code of Atom to confirm it. The Atom team just did not describe this detail in the paper, but have the details in the opensourced code.
- Using different bit-width for different layers is not new, e.g., MixLLM has already proposed this idea (even more fine-grained) and an end-to-end solution. The claim in this paper is not correct: "Unlike prior methods that apply uniform or layer-wise fixed precision".
- Does this paper have any noval technique for microscaling format? It seems this paper only has utilized this data format, but does not have new contribution besides just using it.

---

### Official Review · Reviewer_DufF · 2025-11-01

**Soundness:** 3
**Presentation:** 2
**Contribution:** 2
**Rating:** 2
**Confidence:** 4

**Summary:**

The paper proposes MXSens, a training-free, sensitivity-guided mixed-precision quantization framework designed for microscaling formats (MXINT). The approach combines column-wise (using hessian) and layer-wise (using model error after quantization) sensitivity  to allocate 4, 6, and 8-bit precisions efficiently, enabling compatibility with hardware accelerators that support block-wise scaling.

**Strengths:**

- The paper is easy to follow, well-structured, and includes meaningful ablations on bitwidth configurations.
- **Fine-grained, sensitivity-aware allocation:** Using Hessian-based sensitivity at both column and layer levels is methodologically sound and yields strong empirical performance.
- **Controlled precision budget:** formulating a fixed average bitwidth while improving accuracy is a useful engineering contribution.

**Weaknesses:**

### Weaknesses & Questions

1. **Limited novelty of sensitivity metric:**

    The use of the Hessian diagonal for quantization sensitivity has been well explored in works like **HAWQ-V2** and **APTQ**. The novelty here mainly lies in its adaptation to microscaling rather than the metric itself. The paper should clarify this distinction explicitly.

2. **Missing baselines:**

    Discussion and comparisons exclude recent methods such as **SpinQuant**, **FlatQuant**, **OSTQuant**, **KurTail and DartQuant**.

3. **Lack of comparison with more recent mixed-precision methods:**

    State-of-the-art approaches like **ResQ** show significant gains over **QUIK**. Including at least ResQ would strengthen claims of superiority.

4. **lack of ablation study on design parameters:**

    The hard-coded choice of “top 32 columns per layer → MXINT8” is only loosely justified by block size. An ablation on this or exploration of adaptive thresholds would be interesting.

5. **Hardware feasibility concerns:**

    The paper assumes minimal overhead when columns have heterogeneous bitwidths (4/6/8). How are such columns grouped or stored to avoid bandwidth and alignment penalties?

    To better clarify, different columns could have different number of bit(4,6,8 bits). Isn’t it necessary for all columns with the same bitwidth to be grouped together for the hardware implementation? Further discussion or empirical hardware profiling is needed

6. **Rotation incompatibility not well explained:**

    The claim that rotation “hurts” MXINT quantization due to group-wise asymmetry is mentioned but not analyzed experimentally and at least intuitively in worse case, it should have the same performances as int4. A brief explanation or intuitive justification would clarify this point.

- HAWQ-V2: Hessian Aware trace-Weighted Quantization of Neural Networks, NeurIPS 2020.
- APTQ: Attention-aware Post-Training Mixed-Precision Quantization for Large Language Models, (arXiv preprint) 2024
- SpinQuant: LLM Quantization with Learned Rotations, arXiv/I CLR (pre-print) 2024
- FlatQuant: Flatness Matters for LLM Quantization, ICML 2025
- ResQ: Mixed-Precision Quantization of Large Language Models with Low-Rank Residuals, ICML 2025
- KurTail: Kurtosis-based LLM Quantization, ICLR Workshop (SLLM) 2025
- OSTQuant: Refining Large Language Model Quantization with Orthogonal and Scaling Transformations for Better Distribution Fitting, (arXiv/venue) 2025
- DartQuant: Efficient Rotational Distribution Calibration for LLM Quantization, NeurIPS 2025

**Questions:**

Check weakness

---

### Official Review · Reviewer_6VfQ · 2025-11-21

**Soundness:** 3
**Presentation:** 3
**Contribution:** 2
**Rating:** 4
**Confidence:** 3

**Summary:**

The paper shows that outlier magnitudes vary across columns and layers, and that these differences substantially affect MXINT4 quantization.

**Strengths:**

MXSens consistently outperforms QuaRot, RRS, Atom, and QUIK acrosss LLaMA-2/3, Qwen1.5, and Mistral (Tables 1–2, pages 7–8).
Particularly strong at W4A4KV4, where rotation-based methods typically struggle.

**Weaknesses:**

Algorithm 1 requires accumulating a full diagonal Hessian (page 4). For large models (70B), this is costly and memory-intensive, but the paper does not quantify runtime overhead or provide approximation strategies.

The choice of “top 32 columns → 8-bit” is based on MXINT block size, but: No analysis shows whether always using one high-precision block per layer is optimal.

Work focuses on MXINT, but Figure 1 & 2 show rotation incompatibility mostly for MXFP (AMXFP). Missing discussion on whether triplet quantization generalizes naturally to MXFP.

KV-cache quantization is included (page 7), but: No analysis of how sensitivity is computed for cache entries. Softmax outputs are often problematic; discuss how MXSens affects them.

**Questions:**

See weakness

---

### Meta-Review · Area_Chair_RfWa · 2025-12-21

**Summary:**

The paper under review introduces MXSens, a training-free, sensitivity-guided mixed-precision quantization framework for large language models that leverages column- and layer-wise Hessian estimates to assign variable precision formats. The work aims to address the inherent challenge that outlier magnitudes vary significantly across layers and columns, affecting mixed-precision quantization, particularly under the MXINT format. The authors argue that by detecting and treating sensitive columns with higher precision, overall quantization performance can be improved, as demonstrated by experimental results that show gains over prior rotation-based methods.

**Reviewer Concerns:**

However, the reviewers raised several critical concerns regarding both the methodological novelty and the practical feasibility of the approach. One recurring critique is that the use of the Hessian diagonal as a sensitivity metric has been extensively explored in prior work, and many of the techniques claimed as contributions might closely resemble those proposed in earlier methods such as Atom, HAWQ-V2, and Slim-LLM. The reviewers also noted that the paper omits or inadequately discusses key hardware compatibility challenges, such as the costly accumulation of full diagonal Hessians for large models and the potential runtime overheads when accommodating heterogeneous bitwidths. In addition, insufficient comparisons with recent state-of-the-art baselines and an underdeveloped analysis of certain hyperparameter choices—such as the fixed assignment of high-precision blocks—further undermine the paper’s impact.

**Reviewer Scores:**

Based on the reviewers' evaluations, the consensus is that the submission falls short of the acceptance threshold. While some aspects of the approach are well-motivated and the presentation is clear, the collective concerns about limited novelty, missing baseline comparisons, and unresolved hardware feasibility issues lead to a negative overall assessment. Consequently, the paper is recommended for rejection in its current form.

---

### Decision · Program_Chairs · 2026-01-26

Reject